# Influence of N-Glycosylation on Virus–Host Interactions in *Halorubrum lacusprofundi*

**DOI:** 10.3390/v15071469

**Published:** 2023-06-28

**Authors:** L. Johanna Gebhard, Zlata Vershinin, Tomás Alarcón-Schumacher, Jerry Eichler, Susanne Erdmann

**Affiliations:** 1Max Planck Institute for Marine Microbiology, Archaeal Virology, 28359 Bremen, Germany; 2Department of Life Sciences, Ben-Gurion University of the Negev, Beersheva 84105, Israel

**Keywords:** archaea, archaeal virus, haloarchaea, protein glycosylation, virus

## Abstract

N-glycosylation is a post-translational modification of proteins that occurs across all three domains of life. In Archaea, N-glycosylation is crucial for cell stability and motility, but importantly also has significant implications for virus–host interactions. While some archaeal viruses present glycosylated proteins or interact with glycosylated host proteins, the direct influence of N-glycosylation on archaeal virus–host interactions remains to be elucidated. In this study, we generated an N-glycosylation-deficient mutant of *Halorubrum lacusprofundi*, a halophilic archaeon commonly used to study cold adaptation, and examined the impact of compromised N-glycosylation on the infection dynamics of two very diverse viruses. While compromised N-glycosylation had no influence on the life cycle of the head-tailed virus HRTV-DL1, we observed a significant effect on membrane-containing virus HFPV-1. Both intracellular genome numbers and extracellular virus particle numbers of HFPV-1 were increased in the mutant strain, which we attribute to instability of the surface-layer which builds the protein envelope of the cell. When testing the impact of compromised N-glycosylation on the life cycle of plasmid vesicles, specialized membrane vesicles that transfer a plasmid between host cells, we determined that plasmid vesicle stability is strongly dependent on the host glycosylation machinery. Our study thus provides important insight into the role of N-glycosylation in virus–host interactions in Archaea, while pointing to how this influence strongly differs amongst various viruses and virus-like elements.

## 1. Introduction

Protein glycosylation is a post-translational modification involving the attachment of sugar multimers, or glycans, to specific amino acid residues within a protein. N-glycosylation refers to the attachment of glycans to the nitrogen atom of the amino acid asparagine (Asn), while O-glycosylation involves the attachment of glycans to the oxygen atom of the amino acid serine or threonine. Long thought to be restricted to Eukarya, it is now clear that Archaea are also capable of protein glycosylation. Detailed understanding of N-glycosylation pathways has been obtained for some archaeal species [1], and genome analysis suggests that N-glycosylation is an almost universal trait of Archaea [2,3,4,5,6]. In contrast, O-glycosylation has so far only been observed in a few species, and very little is known of how Archaea perform this modification [1].

In halophilic Archaea (haloarchaea), Archaea that thrive in environments characterized by high salt concentrations, N-glycosylation is important for protein folding, stability, and function [2,7,8,9]. For example, N-glycosylation of the surface (S)-layer glycoprotein, comprising the protein envelope of haloarchaeal cells, is important for cell stability. Additionally, glycan-based modification of haloarchaeal proteins was shown to provide unique markers for pair recognition of mating cells [10,11], for virus–host recognition [12,13] and for virus evasion [11].

In haloarchaea, the N-glycosylation pathway has been extensively studied in the model organism *Haloferax volcanii*. Here, four glycosyltransferases sequentially add four soluble monosaccharides onto a dolichol phosphate carrier on the cytoplasmic side of the plasma membrane [14,15,16,17]. The lipid-linked tetrasaccharide is then flipped to face the extracellular environment, at which point the oligosaccharyltransferase AglB transfers the glycan to selected Asn residues of target proteins [18,19]. Therefore, AglB, with homologues in Eukaryotes and Bacteria (i.e., Stt3 and PglB), can be considered the central enzyme of archaeal N-glycosylation pathways [20]. The fifth and final sugar residue of the pentasaccharide N-linked to *Hfx. volcanii* glycoproteins is attached to a distinct dolichol phosphate carrier on the cytoplasmic side of the membrane. This sugar-charged lipid is translocated across the membrane, and the sugar is added to the glycoprotein-bound tetrasaccharide [16,21,22].

While the general mechanism of N-glycosylation is shared across Archaea, the composition of the N-linked glycans generated is highly diverse, even among species or strains of the same genus [4,11,23]. Such diversity in glycans N-linked to proteins comprising the S-layer and cell appendages has long been considered to play a major role in host-recognition by archaeal viruses.

Accordingly, numerous studies have described interactions of archaeal viruses with highly glycosylated S-layer glycoproteins both in haloarchaea and other archaeal groups. The S-layer of *Halorubrum* sp. SS7-4 was found to trigger fusion of the membrane enveloped Halorubrum Pleomorphic Virus-6 (HRPV-6) with the host cell via the virus spike protein, an interaction that is dependent on the specific host S-layer [24]. The S-layer protein has also been identified as a potential receptor for several archaeal head-tailed viruses [25,26,27] infecting haloarchaea. Archaeal head-tailed viruses do not contain membrane lipids and exit host cells by lysis. Furthermore, partial removal of the highly N-glycosylated S-layer decreased the efficiency of viral infection of the thermophilic archaeon *Sulfolobus solfataricus* (Thermoproteota) by the lipid enveloped Sulfolobus Spindle-shaped Virus 1 (SSV1) [28], that has been shown to exit host cells by budding [29]. Nevertheless, the influence of S-layer protein glycosylation on these interactions is not yet understood.

Conversely, a number of archaeal viruses contain glycosylated proteins. Three of the capsid proteins of SSV1 were found to be glycosylated [30] and are likely required for host attachment and particle stability in the face of high temperatures and acidity. The tail architecture of SSV19, another *Sulfolobus*-targeting virus containing host lipids, includes two highly glycosylated proteins (VP4 and B210). VP4 exhibits a putative glycoside hydrolase core that has been proposed to degrade host S-layer glycans upon binding [31]. The virions of another membrane-enveloped virus infecting hyperthermophilic *Thermoproteota*, Thermoproteus Spherical Piliferous Virus 1 (TSPV1) possess unusual filaments, and glycosylation is thought to contribute to both filament stability at high temperatures and host recognition [32]. Proteins of the lipid-enveloped haloarchaeal virus Halorubrum Pleomorphic Virus 1 (HRPV-1), were shown to be modified by the same glycan as cell surface proteins of its host *Halorubrum* sp. PV6 [12,13]. However, for the majority of archaeal viruses, it remains unknown whether viruses utilize the host N-glycosylation machinery for glycosylation of their proteins. At the same time, a few archaeal viruses encode putative glycosyltransferases [33,34,35,36,37,38], suggesting that there are viruses that could, at least partially, circumvent the host glycosylation machinery. Thus, in most cases, disruption of the N-glycosylation machinery of the host should also affect viral glycosylation, leading to severe effects on virus–host interactions.

To investigate the effect of N-glycosylation on the infection dynamics of archaeal viruses, we generated a *Halorubrum lacusprofundi* mutant strain lacking *Hlac_1062*, annotated as encoding the oligosaccharyltransferase AglB [4,6]. While the composition of glycans decorating *Hrr. lacusprofundi* glycoproteins is currently unknown, cells lacking AglB are expected to be compromised in terms of their ability to perform N-glycosylation. Accordingly, the life cycle of two different viruses and a virus-like element was analyzed and compared between *Hrr. lacusprofundi* parent and N-glycosylation-deficient mutant strains. Furthermore, we tested the impact of N-glycosylation on virus particle stability at different NaCl concentrations. Halorubrum Tailed Virus DL1 (HRTV-DL1) is an archaeal head-tailed virus that was shown to use the S-layer glycoprotein as its primary receptor [26]. Haloferax Pleomorphic Virus 1 (HFPV-1), a membrane-enveloped pleolipovirus [39,40], exhibits a glycosylated spike protein similar to HRPV-1 [12,13]. Finally, plasmid vesicles (PVs), also referred to as plasmidions [41,42], are vesicle-like structures that are morphologically distinct from extracellular vesicles (EVs) [41,43]. PVs contain unique plasmid-encoded proteins which form part of the vesicle coat and disseminate an enclosed plasmid. As such, PVs have been proposed to represent potential evolutionary precursors of some viruses. Hence, comparing the importance of N-glycosylation for host-interactions of true viruses and PVs could help determine the position of PVs within or in proximity to the virosphere [44].

## 2. Materials and Methods

### 2.1. Strains and Cultivation Conditions

All strains solely referred to by their strain name belong to the species *Halorubrum lacusprofundi*. ACAM34_UNSW∆*pyrE2* was grown in modified DBCM2 medium, as previously described [45,46] (180 g/L NaCl, 25 g/L MgCl_2_, 29 g/L MgSO_4_·7 H_2_O, 5.8 g/L KCl, 0.3 g/L peptone, 0.05 g/L yeast extract, 0.006 M CaCl_2_, 2 mL/L K_2_HPO_4_ buffer [47], 0.11 % (*w*/*v*) sodium pyruvate, 0.005 M NH_4_Cl, 1 mL/L SL10 trace elements solution [47] and 3 mL/L vitamin 10 solution [47]), supplemented with 50 µg/mL uracil. Selection medium (DBCM2−) [46] was supplemented with 50 µg/mL uracil, 50 µg/mL 5-fluoroorotic acid (5’-FOA) or with indicated concentrations of pravastatin when needed. To assess the sensitivity of the strains to changing salt concentrations, biological triplicates of cultures were grown in DBCM2+ (DBCM2 with 1 g/L peptone and 0.5 g/L yeast extract), supplemented with 50 µg/mL uracil, and varying NaCl concentrations (100 g/L = 1.7 M, 120 g/L = 2.05 M, 240 g/L = 4.1 M and 250 g/L = 4.3 M, compared to 180 g/L = 3.1 M). During infection experiments and particle stability assays, wild-type ACAM34_UNSW, ACAM34_UNSW∆*pyrE2* and ACAM34_UNSW∆*pyrE2*∆*aglB* were each grown in DBCM2+ medium supplemented with 50 µg/mL uracil. All liquid cultures were grown at 28 °C, with shaking at 120 rpm, rather than at the optimal growth temperature (37 °C, [48]) so as to delay biofilm formation during the stationary phase. Growth on solid media was at 37 °C.

### 2.2. Deletion of Hlac_1062 (aglB)

We targeted *Hlac_1062*, annotated as encoding the oligosaccharyltransferase AglB [4], for gene deletion. The ACAM34_UNSW∆*pyrE2* strain [46] was used to generate the ∆*pyrE2*∆*aglB* deletion strain, using a pop-in/pop-out approach, as described previously [46]. Briefly, approximately 500 bp-long upstream and downstream flanking regions of the *aglB* gene were PCR-amplified (primers used are listed in Appendix A) and inserted into plasmid pTA131_*hmgA* [46,49] using *Eco*RI and *Hind*III restriction sites. The ACAM34_UNSW∆*pyrE2* strain was transformed with the generated plasmid and selected on DBCM2− plates, prepared with DBCM2− medium containing 7.5 µg/mL pravastatin, together with 15 g/L bacteriological agar (OxoidLtd, Basingstoke, United Kingdom). Pravastatin-resistant colonies were re-plated for additional selection on DBCM2− plates with increasing concentrations of pravastatin (10, 15, 15, 30, 45 and 60 µg/mL) over several generations. For the pop-out step, colonies that grew on 60 µg/mL pravastatin plates were selected on plates containing 50 µg/mL 5’-FOA and 50 µg/mL uracil. Finally, selected colonies were screened for *aglB* deletion by PCR and validated by qPCR (primers indicated in Appendix A).

### 2.3. Proteinase K Digestion of the S-Layer Glycoprotein

The effect of *aglB* deletion on the stability of the S-layer was measured by susceptibility to protease treatment compared to the parent strain. Aliquots of *Hrr. lacusprofundi* cultures (three biological replicates per strain) in late exponential growth (OD_600_ ≥ 0.8) were harvested and resuspended in DBCM2 salt solution to an adjusted OD_600_ of 1.0 in 1.2 mL in replicates for both the parent and deletion strains. Proteinase K was added to a final concentration of 0.2 mg/mL and samples were incubated at 37 °C. Aliquots (150 µL) were removed immediately after the addition of proteinase K (0 min) and at several time points until 120 min of incubation. Trichloroacetic acid was added directly to each aliquot (final concentration 15 %), after which samples were incubated at 4 °C for 30 min, centrifuged at 10,000× *g* and 4 °C for 15 min and washed with 1 mL of ice-cold acetone. Acetone was removed from the samples and dried pellets were resuspended in 30 µL (∆*aglB*) or 38 µL (parent) of a mixture containing 1 part sample buffer (0.2 g SDS, 1 mg Bromphenol Blue, 0.78 mL glycerol, 0.2 mL 0.5 M Tris pH 6.8 and 0.155 g DTT per 10 mL) and 5 parts 125 mM Tris-HCl pH 8.0. This difference in volume was chosen to correct for the difference in protein concentration between strains of approx. 0.8:1.0 (∆*aglB*:parent), as determined using Pierce BCA Protein Assay Kit (Thermo Fisher Scientific, Waltham, MA, USA) following the manufacturer’s instructions. Finally, 30 µL portions of each sample were separated by SDS-PAGE (8% acrylamide, Tris-HCl pH 8.8), and stained with Coomassie blue.

### 2.4. Quantification of Extracellular Vesicles

Extracellular vesicles (EVs) were quantified based on a protocol described by Mills et al. [43]. In brief, EVs were isolated from the culture supernatant (*n* = 3 for each strain) via centrifugation (45 min, 4500× *g*) and precipitated from the culture supernatant with PEG_6000_. Precipitated vesicles were resuspended in DBCM2 salt solution [26], filtered through syringe-top filters (0.8, 0.45 and 2× 0.2 µm in sequence) and stained with 500 nM MitoTracker Green FM (Invitrogen) for 30 min. Subsequently, EVs were precipitated after overnight incubation with PEG_6000_ (final concentration of 10%) via centrifugation at 20,000× *g* for 40 min. After removal of the supernatant, the precipitates were resuspended in 200 μL DBCM2 salt solution and fluorescence was measured in a DeNovix, DS-11 FX+ spectrophotometer at emission wavelengths between 514–567 nm, with blue excitation at 470 nm. For transmission electron microscopy (TEM), purified EVs were adsorbed onto carbon-coated copper grids for 5 min, stained with 2% (*w*/*v* in ddH_2_O) uranyl acetate for 1 min and imaged with a JEM2100 Plus TEM at 200 kV acceleration voltage.

### 2.5. Isolation of PVs and Viruses, Infection and Growth Experiments

Infection experiments were performed in biological triplicates using the ACAM34_UNSW∆*pyrE2* (parent) and ∆*pyrE2*∆*aglB* (∆*aglB* mutant) strains. The head-tailed virus HRTV-DL1 was obtained from infected *Hrr. lacusprofundi* ACAM34_UNSW cultures [26]. *Hrr. lacusprofundi* cultures were infected with a multiplicity of infection (MOI) of 10 from a 1.64 * 10^12^ plaque-forming units/mL stock in DBCM2 salt solution. Fresh HFPV-1 stocks were obtained from infected *Hfx. volcanii* cultures following the protocol described by Alarcón-Schumacher et al. [39]. PVs containing the pR1SE plasmid were isolated from *Hrr. lacusprofundi* DL18 [41]. Liquid cultures were inoculated from glycerol stocks, grown until OD_600_ reached approx. 0.8 and subsequently diluted into fresh DBCM2+ medium to OD_600_ 0.05. This process was repeated three times, at which point the cultures were diluted to OD_600_ 0.05 and supplemented with 30 mL of 10× YPC solution [47] to induce increased PV production [41]. Cells were harvested in early stationary phase (OD_600_ 1.3 − 1.6) at 4500× *g* for 45 min at room temperature (RT). PVs were precipitated with 10 % PEG_6000_ (final concentration) overnight at 4 °C, harvested at 30,000× *g* for 45 min at 4 °C (JA-14, Beckman-Coulter) and the precipitate was resuspended in 7 mL of DBCM2 salt solution. The PV preparation was centrifuged for 10 min at 10,000× *g*, and the supernatant was filtered once through a 0.45 µm syringe filter and three times through 0.2 µm filters to remove remaining cells. Since the number of plasmids per PV has not yet been determined and because PVs do not induce plaque formation, precise quantification of PVs in suspension is not possible. Infection was, therefore, performed with a plasmid copy number:host cell ratio of approximately 75:1.

Infection of ACAM34_UNSW∆*pyrE2* and ∆*pyrE2*∆*aglB* was performed as described by Alarcón-Schumacher et al. [39]. Briefly, cultures were diluted three times from mid-exponential growth to OD_600_ 0.05, harvested at mid-exponential phase by centrifugation (4500× *g* for 45 min at RT), resuspended in 1 mL medium and subsequently incubated with the virus or PV stock in DBCM2 salt solution, followed by removal of unbound particles by washing after 2 h. For HRTV-DL1, free virus particles were not removed from infected cell suspensions after the 2 h infection step, with the mixture being directly transferred into the cultures, as described [26]. Each strain (*n* = 2) was infected in biological triplicates for each infectious element (*n* = 3); biological triplicates of uninfected controls (*n* = 3) were performed for each infectious agent (total *n* = 36). All cultures were incubated at 28 °C and 120 rpm, and growth was monitored by measuring OD_600_. Aliquots for quantification by qPCR were removed at the time points indicated, covering early infection, exponential growth and the late exponential phase. Two milliliters of culture from each biological replicate was centrifuged at 18,000× *g* for 10 min at RT. The cell pellet was washed twice with DBCM2+ medium, snap frozen with liquid N_2_ and stored at −80 °C until DNA extraction. Virus particles and PVs were precipitated from 1.5 mL (HRTV-DL1 and PVs) or 1 mL (HFPV-1) supernatant with 10% final concentration of PEG_6000_ (as described above) and the resulting pellets were snap-frozen with liquid N_2_ and stored at −80 °C until DNA extraction. For PV infection, extended cultivation was performed by diluting (to OD_600_ 0.05) aliquots into fresh DBCM2+ medium (150 mL) at late exponential growth phase after the final aliquot was removed during the initial cultivation.

### 2.6. Virus and PV Quantification

All DNA extractions were performed with an Isolate II Genomic DNA Kit (Bioline, London. United Kingdom) according to the manufacturer’s instructions. Host, virus, and plasmid genome copy numbers were quantified by qPCR as described previously in technical triplicates for each biological replicate [26,39]. Primer sequences, annealing temperatures and primer concentrations are listed in Appendix A. Genome copy numbers in all samples were quantified with a CFX96 Touch Real-Time PCR machine (Bio-Rad Laboratories, Hercules, CA, USA) and CFX Manager Software in 10 µL reactions with SsoAdvanced Universal SYBR Green Supermix (Bio-Rad, Hercules, CA, USA). The qPCR program consisted of 5 min at 95 °C and 30 s at 95 °C, 30 s at X °C for 35–40 cycles (X, corresponding to the annealing temperature of the respective primer sets, is indicated in Appendix A), with measurements being taken between each cycle. Assays were only considered for quantification when efficiencies were between 95–105% and R^2^ ≥ 0.98. Prior to statistical analysis, normal distribution and homogeneity of variances in sample sets separated by virus and time point were confirmed based on the consensus of a Shapiro–Wilk Test, F-Test, Bartlett’s Test [50], Levene’s Test [51] and a Fligner–Killeen’s Test [52]. The means from parent and ∆*aglB* samples were compared with two-sided, unpaired Student’s *t*-tests. All analysis was performed with R version 4.1.2 [53] and RStudio version 2023.3.0.386 [54].

### 2.7. Glycostaining of Cells and Particles

Periodic acid-Schiff staining following the protocol of Dubray and Bezard [55] was performed on aliquots of *Hrr. lacusprofundi* cells that were separated by SDS-PAGE (8% acrylamide, Tris-HCl pH 8.8). HRTV-DL1 virions and PVs were obtained from the supernatant of infected *Hrr. lacusprofundi* cultures, HFPV-1 from *Hfx. volcanii*, and purified as described above. Aliquots of each infectious agent were separated by SDS-PAGE (8% acrylamide, Tris-HCl pH 8.8) and glycosylated proteins were stained with the Pro-Q Emerald 300 glycoprotein gel and blot stain kit (Invitrogen, Waltham, MA, USA), using periodate-based oxidation of carbohydrate groups according to the manufacturer’s instructions. Total protein content was visualized with Coomassie blue.

### 2.8. Determination of Virus Stability

Cultures of the ACAM34_UNSW∆*pyrE2* and ∆*pyrE2*∆*aglB* strains were each infected with HRTV-DL1 or HFPV-1 in separate experiments. Virions were harvested from the culture supernatants and purified according to the protocols described above. Purified preparations from each strain were precipitated again into three aliquots with PEG_6000_ and resuspended in 0.075 mL DBCM2+ medium prepared with 120 g/L (2.05 M), 180 g/L (3.1 M), or 240 g/L (4.1 M) NaCl. After overnight incubation at room temperature, 1 mL aliquots of the wild type ACAM34_UNSW strain (OD_600_ = 1.0) [26] were infected with viral preparations (biological triplicates per virus), with sterile medium (biological triplicates per virus) serving as the negative control. MOIs were 5 (for ∆*aglB*-strain-produced virions) or 1 (for parent-strain-produced virions) for HRTV-DL1 and 3.5 (for ∆*aglB*-strain-produced virions) or 5.3 (for parent-strain-produced virions) for HFPV-1, respectively. Differences in MOI were caused by disparate yields of viral preparations from infected strains. After incubation (2 h), cells were washed twice with sterile medium and subsequently snap-frozen in liquid nitrogen. Viral copy numbers in DNA extracts of cell pellets were determined via qPCR as described above and compared to copy numbers of the *Hrr. lacusprofundi* chromosome. The infection efficiency was calculated as a percentage. Intracellular viral genome copy numbers per ng of DNA from each biological replicate were multiplied by the total DNA content (in ng) of the respective sample and subsequently divided by the total number of viral genome copy numbers added to each replicate during infection. Homogeneity of variances was confirmed based on the consensus of Shapiro–Wilk, Bartlett’s [50], Levene’s [51] and Fligner–Killeen’s tests [52]. The effect of NaCl concentration on infection efficiency was determined with two-sided, unpaired student’s *t*-tests, per virus and per strain. All analysis was performed with R version 4.1.2 [53] and RStudio version 2023.3.0.386 [54].

## 3. Results

### 3.1. Deletion of Hrr. lacusprofundi aglB Leads to Growth Deficiencies at Varying Salt Concentrations and S-Layer Instability

A putative glycosylation cluster was previously identified in *Hrr. lacusprofundi* (*Hlac_1062* to _*1075* [4]), including the archaeal oligosaccharyltransferase *aglB* within the vicinity of several genes annotated as encoding predicted glycosyltransferases, putative sugar epimerases and one putative flippase. In order to manipulate the *Hrr. lacusprofundi* N-glycosylation pathway, we generated a knockout strain for *Hlac_1062* (*aglB*). Deletion of *aglB* was confirmed by qPCR (Figure 1A), while glycoprotein staining confirmed strongly decreased glycosylation of the S-layer glycoprotein in the mutant strain (Appendix A). In *Hfx. volcanii*, deletion of *aglB* leads to instability and increased release of the S-layer glycoprotein [18]. Accordingly, protease treatment of *Hrr. lacusprofundi* parent and mutant cells revealed increased protease sensitivity of the mutants’ S-layer glycoprotein, reflecting a change in S-layer glycoprotein conformation and subsequently, increased instability of the S-layer in the *aglB*-deletion strain (Figure 1B). The ∆*aglB* strain (ACAM34_UNSW∆*pyrE2*∆*aglB*) either grew to the same or to a slightly higher optical density (600 nm) than the parent strain (ACAM34_UNSW∆*pyrE2*) in standard DBCM2 media containing a NaCl concentration of 180 g/L (3.1 M) (Figure 1C, Appendix A). However, growth of the ∆*aglB* strain was negatively impacted both at a high NaCl concentration of 240 g/L (4.1 M) and in low-salt medium containing 120 g/L NaCl (2.05 M) (Figure 1B). Unlike the parent strain, the mutant strain was unable to grow in 250 g/L NaCl medium (4.3 M) and both strains struggled at 100 g/L NaCl (1.7 M) (Appendix A). These observations mirror findings previously reported for *Hfx. volcanii* ∆*aglB* cells [18].

S-layer instability was suggested to be responsible for the rise of extracellular vesicle (EV, [56,57,58]) production in *Hfx. volcanii* [43]. We, therefore, assessed EV production in the *Hrr. lacusprofundi* parent and ∆*aglB* mutant strain. EVs could be isolated from both strains (Figure 2A,B). Quantification of EVs using a fluorescence-based assay (see Methods) revealed increased vesicle production by the ∆*aglB* stain, relative to the parent strain (Figure 2C), confirming reduced stability of the S-layer.

### 3.2. The Life cycle of the Lytic Virus HRTV-DL1 Is not Influenced by Changes in Host N-Glycosylation

HRTV-DL1 is an archaeal tailed virus that causes a lytic infection in ACAM34_UNSW and was shown to use the host cell S-layer for attachment [26] (Table 1). The major capsid protein that builds the virus particle presents three N-glycosylation motifs or sequons (i.e., NXS/T, where X is any residue but Pro [59]), while some of the other structural proteins that form the neck or tail of the virion also contain (<5) sequons. The protein encoded by ORF24, suggested to play a role in host attachment, presents nine such motifs and may thus be N-glycosylated [26]. Glycoprotein staining of viral proteins revealed two faint bands >72 kDa, possibly including the ORF24 protein, and one band <72 kDa, that could represent the major capsid protein encoded by ORF12 (Appendix A).

Infection with HRTV-DL1 led to the characteristic lysis of both the parent strain and ∆*aglB* mutant cells about 23 h post infection (h.p.i.) (Figure 3A). Viral titers in the culture supernatants strongly increased after lysis, reaching similar maximal values for both strains (3.2*10^+11^ ± 9.7*10^+10^ per mL, *n* = 3, for the parent strain versus 1.35*10^+11^ ± 1.2*10^+11^ per mL, *n* = 3, for the ∆*aglB* mutant) at approximately 40 h.p.i. The intracellular virus genome copy numbers (Appendix A), as well as the virus:host genome ratio (Figure 3B), were not significantly different between the two strains (*p*-values < 0.05). This led us to conclude that the deletion of *aglB* did not have a significant effect on the life cycle of HRTV-DL1.

### 3.3. Intracellular and Extracellular Virus Numbers of the Chronic Virus HFPV-1 Are Significantly Increased in the ∆aglB Mutant

Proteins comprising the HFPV-1 virus particle contain N-glycosylation sequons and protein glycosylation was confirmed by glycostaining of the SDS-PAGE-separated proteins from purified particles (Appendix A). Additionally, no glycosylation-related genes that could compensate for compromised host N-glycosylation were identified on the genome of HFPV-1 (Table 1), suggesting that the impairment of N-glycosylation due to *aglB* deletion would have an effect on the life cycle of HFPV-1.

Infection with the HFPV-1 virus led to a slight growth retardation during early exponential growth (ca. 25 h.p.i.) of both the parent and ∆*aglB* strains (Figure 4A). HFPV-1 causes a productive chronic infection, in which viral particles are released by extrusion or by budding through the cell membrane without lysis of the host [39,60]. Viral titers in the culture supernatant did not show a sharp increase, as observed after lysis of HRTV-DL1-infected cultures, but rather remained within similar orders of magnitude (10^+7^–10^+8^) throughout growth, although titer values did decline over time. Interestingly, we detected significant differences in viral titer between strains, with consistently higher titers being recorded for the ∆*aglB* strain. Maximal viral genome copy numbers per ml were three times higher (*p =* 0.0001; Figure 4A) for the mutant strain (6.42*10^+8^ ± 4.8*10^+7^, *n* = 3) at 44 h.p.i than in the parent strain (1.89*10^+8^ ± 2.4*10^+7^, *n* = 3). Intracellular virus:host genome ratios remained consistently below one for both strains (Figure 4B, Appendix A), yet were higher in the mutant strain with a statistically significant difference (e.g., *p* = 0.0009 at 44 h.p.i), except at the last time point, displaying a consistent pattern with the viral titers in the culture supernatants.

### 3.4. PV Production Is Reduced in the aglB Deletion Strain

PV plasmid-encoded proteins contain N-glycosylation sequons. Protein glycosylation was confirmed by glycostaining of the SDS-PAGE-separated proteins from purified particles (Appendix A). Interestingly, the pR1SEDL18 plasmid (produced by *Hrr. lacusprofundi* strain DL18 [41], used in this study) encodes two predicted glycosyltransferases that have been identified in PVs and host cell membrane preparations ([41], Table 1). These could potentially impact the glycosylation of PVs, yet are unlikely to fully compensate for the lack of host *aglB*.

Infection with PVs (containing plasmid pR1SE) did not result in any major changes in the growth rate of either the parent or ∆*aglB* strains (Figure 5). Infection efficiency of PVs was the same for the ∆*aglB* mutant and the parent strain (average of 6.83% ± 1.4% for ∆*aglB* strain and 3.45% ± 1.92% for parent strain), indicating that infection is independent of N-glycosylation of the host cell envelope proteins and S-layer stability. The plasmid could be detected at 20.25 h.p.i., with the titer reaching a maximum value of 7.31*10^+8^ (± 1.5*10^+8^, *n* = 3) in culture supernatants of the parent strain, confirming production of PVs. Maximum plasmid titers per ml reached higher values in parent cultures (7.87*10^+8^ ± 3.15*10^+8^, *n* = 3; at 43 h.p.i than in the ∆*aglB* strain (Figure 5A). Intracellular copy numbers of the infectious plasmid pR1SE reached a maximum of 9.8*10^+5^*ng DNA^−1^ (± 6.9*10^+4^, *n* = 3) at 20.25 h.p.i. for the parent strain and 7.3*10^+5^*ng DNA^−1^ (± 1.54*10^+5^, *n* = 3) at 20.25 h.p.i. for the ∆*aglB* strain (Appendix A). Plasmid:host ratios reached up to 7.6 (20.25 h.p.i.) and 3.6 (20.25 h.p.i) in the mutant and parent strains, respectively, with statistically significant differences between strains being noted during early infection (20.25 h.p.i., *p* = 0.0002) and exponential growth (43 h.p.i., *p* = 0.03*). Interestingly, despite higher plasmid:host ratios in the ∆*aglB* mutant at 20.25 h.p.i., the extracellular titer remained significantly lower relative to the parent strain. However, reduction of PV titers in the ∆*aglB* mutant was at most only fourfold, suggesting that the impacted N-glycosylation of PV proteins did not prevent the formation of PVs. Additionally, the release of both extracellular vesicles and enveloped virus HFPV-1 was increased in the ∆*aglB* mutant, demonstrating that release of membranous structures was not hindered. Therefore, we suggest that the reduction of extracellular plasmid numbers is due to reduced stability of PV particles produced by the ∆*aglB* mutant.

To test whether PV production was consistent over longer-term infections, we extended the growth of the plasmid pR1SE-infected cultures. Interestingly, while plasmid: host ratios increased over time in both strains, extracellular plasmid numbers did not increase accordingly (Figure 5B,C), indicating that PV production is not coupled to intracellular plasmid copy numbers. Extracellular plasmid copy numbers remained consistently lower in the ∆*aglB* mutant, confirming the negative effect of N-glycosylation deficiency on PV stability.

### 3.5. Impaired Glycosylation has an Impact on the Stability of HFPV-1

Since PV production was reduced in the ∆*aglB* mutant, we concluded that PVs produced in the ∆*aglB* mutant are likely less stable, and that the same could possibly apply to virus particles. We had previously determined that the ∆*aglB* mutant is sensitive to changes in NaCl concentration (Figure 1C). Therefore, we exposed virus particles produced in the parent and ∆*aglB* strains to different salt concentrations to test particle stability. We calculated infection efficiency of the particles at 2 h.p.i as a measure of particle stability.

For HRTV-DL1 particles (Figure 6A), the salt concentration had a significant effect on the infectivity of particles produced in both strains, with high salt promoting particle stability (Figure 6). However, no significant difference was observed when comparing the effect between strains. Varying salt concentration also had a statistically significant effect on HFPV-1 particles when produced in the parent strain (Figure 6B). Particles appeared to be more stable in physiological and high-salt conditions when compared to low-salt conditions (*p* = 0.039 and *p* = 0.017 respectively). In contrast, particles produced in the ∆*aglB* strain only showed increased stability in high-salt conditions when compared to low-salt conditions. This indicates that HFPV-1 particle stability is altered when N-glycosylation is impaired.

## 4. Discussion

In this study, we generated an in-frame deletion mutant of *Hlac_1062*, corresponding to the archaeal oligosaccharyltransferase-encoding *aglB* gene in *Hrr. lacusprofundi*. The oligosaccharyltransferase is considered to be the central enzyme of N-glycosylation pathways in Eukaryotes, Bacteria and Archaea [20]. To date, every archaeal species with *aglB* deleted has shown a loss of N-glycosylation [1,18,61,62], except for *Sulfolobus acidocaldarius*, where *aglB* is essential and could not be deleted [1]. Yet, in *Hfx. volcanii*, evidence for the existence of a second N-linked glycan generated in low-salt conditions and added to protein targets in an AglB-independent manner has been presented [63,64]. *Hrr. lacusprofundi* proteins may be additionally modified by O-linked glycans, as previously described for other haloarchaea [65,66,67,68], although genes involved in the assembly of this glycan have not yet been identified. Thus while, deletion of *Hrr. lacusprofundi aglB* could theoretically not lead to a complete loss of N-glycosylation, we nonetheless showed that N-glycosylation was strongly impacted by such deletion (Appendix A). Moreover, the *Hrr. lacusprofundi aglB* mutant showed growth deficiencies in medium containing salt levels below and above the optimum concentration, confirming that N-glycosylation is important for cell stability and adaptability to changing extracellular salt concentrations. Given the increased susceptibility of the S-layer glycoprotein to proteinase K treatment (Figure 1B) and the increased production of extracellular vesicles (Figure 2C) [10,43] in the deletion strain, we concluded that a reduced stability of the S-layer resulted in compromised N-glycosylation, suggesting that such post-translational modification is very important for the function of the *Hrr. lacusprofundi* S-layer [41,69], as previously shown for both *Hfx. volcanii* [15] and *Halobacterium salinarum* [9]. To gain insight into the relationship between N-glycosylation and viral infection, we used the ∆*aglB* mutant to analyze the importance of this post-translational modification on the life cycle of PVs and two viruses with different lifestyles infecting this species (Table 1).

HRTV-DL1 is a head-tailed virus that exhibits a lytic life cycle, with virus particles being released via cell lysis without incorporating host membrane lipids. Glycosylation was shown for some of the structural proteins, possibly including the major capsid protein. Infection with HRTV-DL1 was not impacted by impaired N-glycosylation, and neither HRTV-DL1 intracellular replication nor virus particle production were affected in the ∆*aglB* mutant. While changes in salt concentrations had strong effects on HRTV-DL1 particle stability in general, with high-salt conditions preserving virion infectivity, there was no evidence for an impact of N-glycosylation on particle stability. The N-glycosylation status of the host receptor, the S-layer glycoprotein [26], does not affect the life cycle of HRTV-DL1 which indicates that neither host recognition nor virus adsorption is dependent on sugar residues on the host cell surface, as has been shown for a number of bacterial tailed viruses. Carbohydrate-binding domains have been widely identified as playing a role in host recognition and attachment of siphophages infecting Gram-positive bacteria [70,71,72,73], where the thick peptidoglycan layer can be considered analogous to the archaeal S-layer. The host ranges of archaeal tailed viruses of the *Caudoviricetes* class vary greatly, even among members of the same family [25]. HRTV-DL1 exhibits a very narrow host range, infecting only a laboratory strain of *Hrr. lacusprofundi,* whereas other archaeal tailed viruses can infect hosts across haloarchaeal genera [25,74,75,76]. Therefore, host recognition and attachment via binding to glycan groups exposed on the host S-layer might well play a crucial role for other archaeal tailed viruses.

In contrast to HRTV-DL1, deletion of *aglB* affected the reproduction of HFPV-1 and plasmid pR1SE. Both infectious agents most likely release particles by budding from the host membrane and probably rely on the host machinery to glycosylate their structural proteins. HFPV-1 was detected at comparatively low copy numbers both within host cells and in the supernatant of the tested strains, which we attribute to the fact that *Hrr. lacusprofundi* is not the optimal host for this virus [39]. Nevertheless, we did detect a significant difference in both intracellular and extracellular virus genome copy numbers, numbers that were significantly higher in the ∆*aglB* mutant. We conclude that HFPV-1 does not rely on host N-glycosylation for interactions with the host cell, similar to HRTV-DL1 [39]. At the same time, destabilization of the S-layer caused by incorporation of non- or incompletely N-glycosylated S-layer glycoproteins may have improved accessibility of the host cell envelope during both virus entry and exit. In a similar way to the increased EV production seen in the ∆*aglB* mutant, the destabilized S-layer seems to allow for increased virus budding.

Interestingly, while a less rigid S-layer appears to be beneficial for continuous virion release through budding, it does not have a significant effect on HRTV-DL1 virion release via cell lysis. This indicates that the mechanism of HRTV-DL1-induced lysis is entirely driven by the virus and is either independent of the condition of the S-layer or includes S-layer destabilization. It has been suggested that HRTV-DL1 uses a glycosyl hydrolase (ORF26) to degrade surface glycans and destabilize the S-layer, while a holin-like protein disrupts membrane integrity and leads to lysis. In contrast, HFPV-1 release seems to benefit from partial instability of the cell envelope. No protease encoded in the genomes of membrane-containing archaeal viruses has been detected to date [77], suggesting that enveloped viruses may either target S-layer stability by other means, such as attacking glycosylation or blocking assembly of the S-layer. Alternatively, they might entirely depend on partial S-layer instability where it naturally occurs, for example, at sites of cell division, when S-layer glycoproteins are naturally released during growth of the host [18], or during C-terminal post-translational modifications of the S-layer glycoprotein at the cell surface, specifically lipid attachment and proteolytic cleavage that could introduce temporary S-layer disruption [5,78,79,80].

While extracellular copy numbers of plasmid pR1SE were significantly reduced in the ∆*aglB* strain, the difference in infection efficiencies between strains was not statistically relevant. Therefore, we conclude that recognition of the host cell and contact of PVs with the host cell are not dependent on interactions with host glycans, as for HRTV-DL1 and HFPV-1. However, extracellular PV numbers appeared to be strongly impacted by the host glycosylation deficiency, an effect we suggest results from reduced particle stability.

We also confirmed the altered stability of HFPV-1 particles produced by the ∆*aglB* mutant upon exposing particles to different salt concentrations. The spike protein of HFPV-1 and the putative vesicle coat protein of PVs are both in all likelihood N-glycosylated [39,41] (Appendix A). Considering the altered particle stability of PVs and HFPV-1, we conclude that both are reliant on the host glycosylation machinery for adding glycans to their structural surface proteins, and that N-glycosylation is crucial for their stability. We assume that reduced stability might mask any potential increase in PV titers caused by instability of the cell envelope that we detect for HFPV-1. Similarly, while we did see a statistically significant increase in HFPV-1 titers in the mutant strain, these are not very high. We suggest that the reduced particle stability of HFPV-1 negatively affected the final titers observed. We conclude that, for both HFPV-1 and PVs, impairment of N-glycosylation has two major effects, namely reduced particle stability and increased particle release due to S-layer instability, while the impact of each effect differs for each of the two elements at the conditions tested.

Finally, this study provided new insight into the nature of the PV life cycle. PV production was already detected 20 h post-infection, similarly to the production of HFPV-1 and HRTV-DL1 particles. The plasmid:host ratio of plasmid pR1SE (up to 15.4) was significantly lower than that of the two viruses (virus:host ratio up to 10^3^ for HFPV-1 in its preferred host *Hfx. volcanii* [39], and up to 297 for HRTV-DL1). However, this ratio was significantly higher than those of haloarchaeal mini-chromosomes or mega-plasmids (>100 kb, copy numbers ≤2) [81,82] and smaller archaeal plasmids (≤10 kb, copy numbers ranging from 6–15) [50,83,84]. Furthermore, plasmid:host ratios of pR1SE increased over time during prolonged infection, in contrast to previously reported data on the dissemination of large conjugative plasmids (~25–45 kb) in the *Sulfolobaceae*, where copy numbers declined over time to reach stable low copy numbers in the long-term [85,86,87]. We suggest that this difference is caused by the dissemination mechanism of plasmid pR1SE, which is more similar to the transmission of viruses rather than the transmission of conjugative plasmids. Thus, this mobile genetic element behaves more similarly to an actively replicating virus than to a conventional plasmid. The rapid replication and dissemination of plasmid pR1SE do not have any apparent negative effect on host fitness, posing further questions as to the nature of the interaction between the host and these plasmid vesicles at the edge of the virosphere [44].

## 5. Conclusions

This study revealed that N-glycosylation is not only crucial for *Hrr. lacusprofundi* cells to adapt to changing environmental conditions, but also for the stability of some viruses and virus-like particles that include glycosylated proteins. At the same time, N-glycosylation does not play a role in host cell recognition and attachment for HRTV-DL1, HFPV-1 and PVs. We propose that destabilization of the S-layer leads to increased entry and exit of membrane-enveloped HFPV-1, indicating that HFPV-1 might depend on naturally occurring gaps in the S-layer for exit and entry. Finally, our study revealed that plasmid pR1SE copy numbers range between those of viruses and similarly sized plasmids, supporting the hypothesis that PVs are an intermediate between plasmids and viruses, potentially representing evolutionary precursors of viruses.

## Figures and Tables

**Figure 1 viruses-15-01469-f001:**
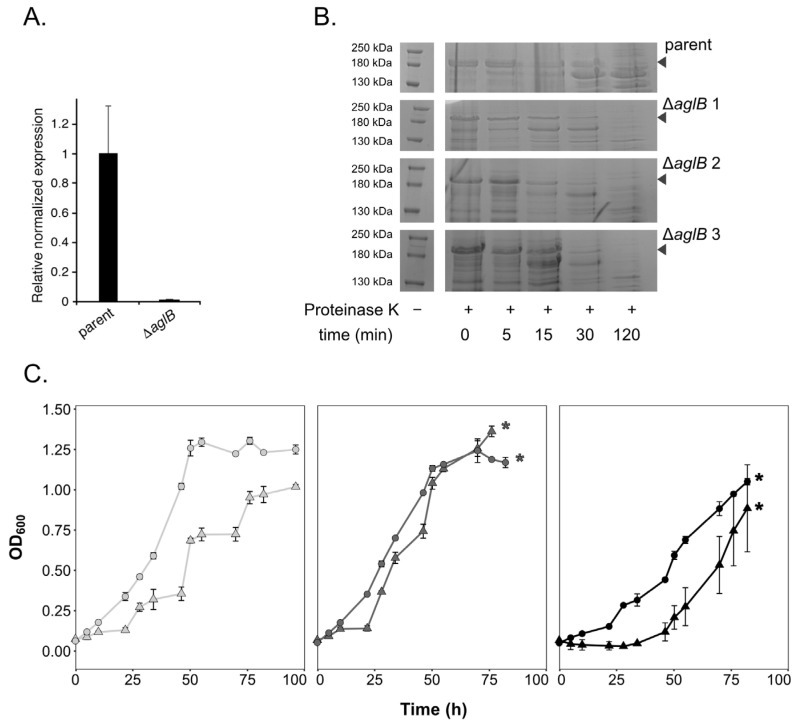
**The *Hrr. lacusprofundi* ∆*aglB* strain shows altered growth and increased protease sensitivity of the S-layer.** (**A**) Validation of *aglB* deletion by qPCR. The relative mRNA expression of *aglB* (*Hlac_1062*) was normalized to that of a housekeeping gene (*16S rRNA*). The values presented are the average of four technical repeats ± standard error of the mean. (**B**) Parent and ∆*aglB* cells were challenged with 0.2 mg/mL proteinase K at 37 °C for the indicated times (min). Precipitated cellular proteins are visualized on 8% acrylamide SDS-PAGE gels for one representative replicate (out of three biological replicates) from the parent strain and three biological replicates from the ∆*aglB* strain. The position of the S-layer glycoprotein is indicated with an arrow. (**C**) The growth of parent (circles) and ∆*aglB* (triangles) strains was monitored in media with NaCl concentrations of 120 g/L (2.05 M, light grey), 180 g/L (3.1 M, dark grey) or 240 g/L (4.1 M, black). Each point represents the average of three biological replicates ± standard deviation of the mean. The data were generated simultaneously for both strains across NaCl concentrations in a separate experiment to the data shown in Appendix A. Asterisks indicate when cultures went into biofilm and OD_600_ could no longer be accurately measured.

**Figure 2 viruses-15-01469-f002:**
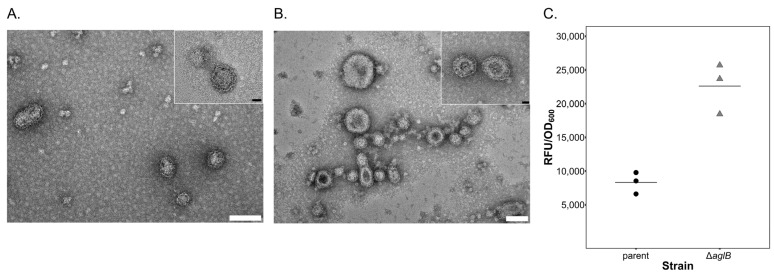
**Extracellular vesicle production is increased upon *aglB* deletion.** Electron micrographs of EVs isolated from the supernatant of the parent strain (**A**) and the ∆*aglB* mutant (**B**). The white scale bars represent 100 nm and the black scale bars in the inserts represent 20 nm. (**C**) EVs in the culture supernatant were quantified by fluorescent labelling (relative fluorescent units, RFUs, normalized to OD_600_) of the parent and ∆*aglB* strains. Individual measurements from three biological replicates are presented, and the average value for each strain is indicated by the horizontal line.

**Figure 3 viruses-15-01469-f003:**
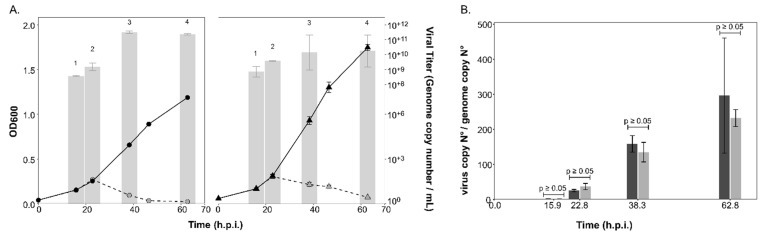
**Growth of the parent and ∆*aglB* strains and virus titers during infection with HRTV-DL1.** (**A**) Growth of uninfected (black, solid line) and infected (grey, dashed line) cultures of *Hrr. lacusprofundi* parent (circles) and ∆*aglB* (triangles) strains over time (h.p.i.). Virus titers in the culture supernatant (genome copy number/mL) are shown as bar graphs in logarithmic scale (log base 10). Data points and bars represent the average of three biological replicates (*n* = 3) ± standard deviation of the mean. Samples were compared with two-sided, unpaired Student’s *t*-tests per time point (1–4), which did not reveal a significant difference between strains (1: *p* = 0.4, 2: *p* = 0.11, 3: *p* = 0.10 and 4: *p* = 0.3). (**B**) HRTV-DL1 genome copy numbers normalized to the level of the host chromosome within cells. Bars represent the average of three biological replicates (*n* = 3) ± standard deviation of the mean for the parent (dark grey) and ∆*aglB* strain (light grey). Samples were compared with two-sided, unpaired Student’s *t*-tests per time point, which did not reveal a significant difference between strains (as indicated above the bars with *p* ≥ 0.05).

**Figure 4 viruses-15-01469-f004:**
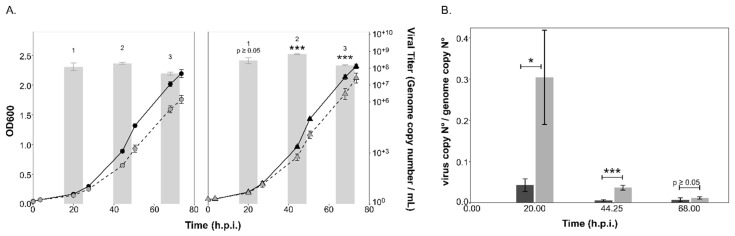
**Growth of the parent and ∆*aglB* strains and virus titers during infection with HFPV-1.** (**A**) Growth of uninfected (black, solid line) and infected (grey, dashed line) cultures of *Hrr. lacusprofundi* parent (circles) and ∆*aglB* (triangles) strains over time (h.p.i.). Virus titers in the culture supernatant (genome copy number/mL) are shown as bar graphs in logarithmic scale (log base 10). Data points and bars represent the average of three biological replicates (*n* = 3) ± standard deviation of the mean. Statistical significance is indicated in the graph for the ∆*aglB* strain, with the following significance codes: ‘***’ for *p* < 0.001, ‘*’ for *p* ≤ 0.05 and *p* ≥ 0.05. (**B**) HFPV-1 genome copy numbers normalized to the level of host chromosome within cells. Bars represent the average of three biological replicates (*n* = 3) ± standard deviation of the mean for the parent (dark grey) and ∆*aglB* strains (light grey). Statistical significance is indicated in the graph using the same significance codes as described above for (**A**).

**Figure 5 viruses-15-01469-f005:**
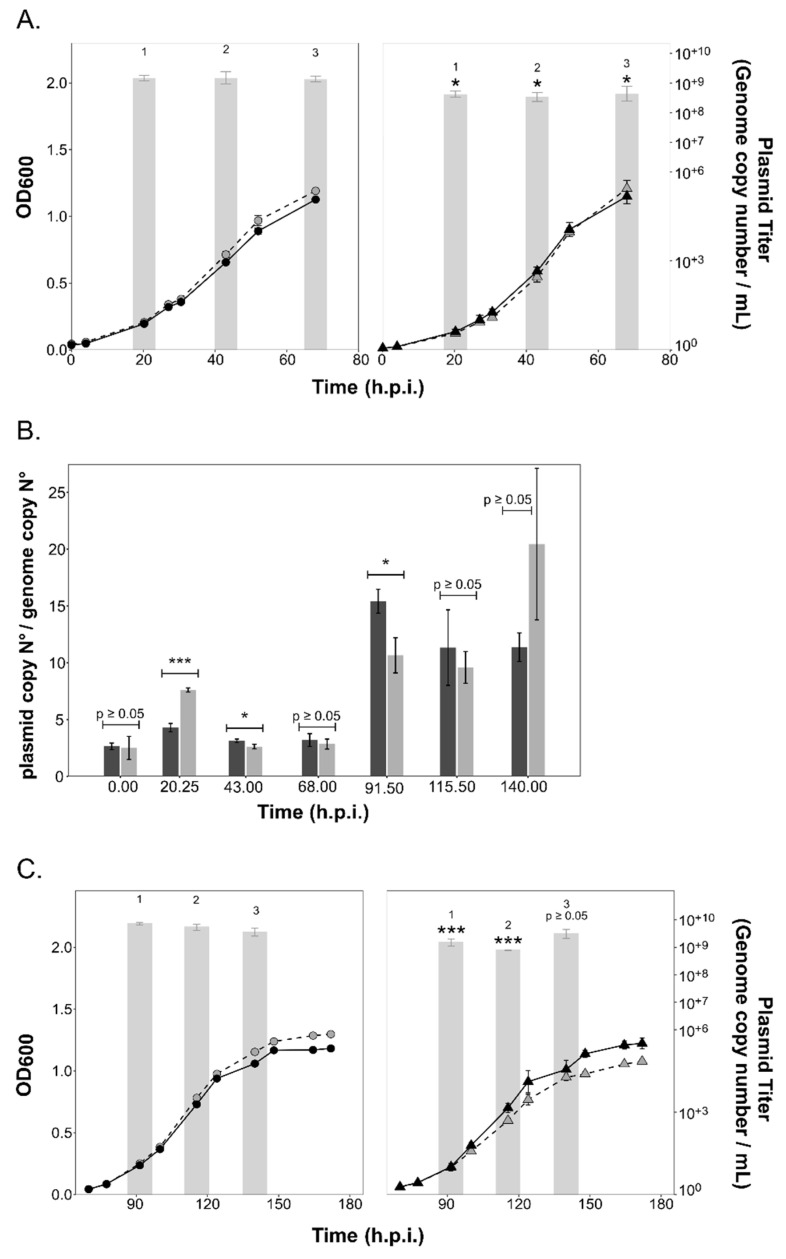
**Growth of the parent and ∆*aglB* strains and plasmid titers during infection with plasmid pR1SE.** (**A**) Growth of uninfected (black, solid line) and infected (grey, dashed line) cultures of *Hrr. lacusprofundi* parent (circles) and ∆*aglB* (triangles) strains over time (0–68 h.p.i.) Plasmid titer of pR1SE in the culture supernatant (genome copy number/mL) are shown as bar graphs in logarithmic scale (log base 10). Data points and bars represent the average of three biological replicates (*n* = 3) ± standard deviation of the mean. Statistical significance is indicated in the graph above the values for the ∆*aglB* strain, with the following significance codes: ‘***’ for *p* < 0.001, ‘*’ for *p* ≤ 0.05 and *p* ≥ 0.05. (**B**) Plasmid pR1SE genome copy numbers normalized to the level of the main host chromosome within cells. Bars represent the average of three biological replicates (*n* = 3) ± standard deviation of the mean for parent (dark grey) and ∆*aglB* (light grey) strains. Statistical significance is indicated in the graph using the same significance codes described above for (**A**). (**C**) Continued growth of cultures after dilution of the cultures shown in (**A**) with the same graphical representation and statistical analysis.

**Figure 6 viruses-15-01469-f006:**
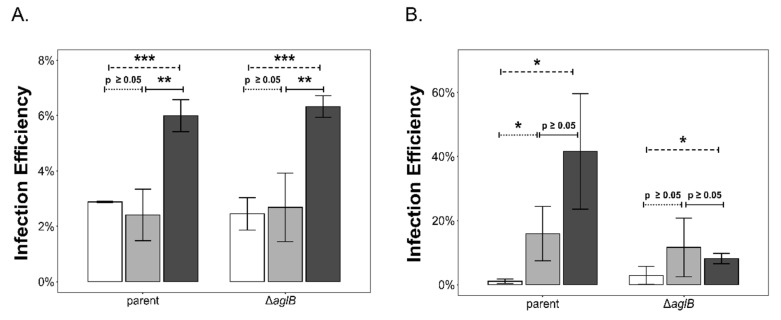
**Deletion of *aglB* negatively affects particle stability of HFPV-1.** Infection efficiency of HRTV-DL1 (**A**) and HFPV-1 (**B**) particles after overnight incubation in medium containing 120 (2.05 M, white), 180 (3.1 M, light grey) or 240 g/L (4.1 M, dark grey) NaCl was tested on wild type *Hrr. lacusprofundi* cells, as a measure of particle stability. Infection efficiency (within 2 h) is given as the percentage of viral genome copy numbers inside the cells divided by total viral copy numbers used for infection per replicate. Bars represent the average of three independent biological replicates (*n* = 3) ± standard deviation of the mean for viruses produced in the parent and ∆*aglB* strains. Infection efficiencies were compared in pairs separately for each strain and virus as indicated in the graph (dotted line = 120 g/L against 180 g/L, dashed line = 120 g/L against 240 g/L, solid line = 180 g/L against 240 g/L). Statistical significance is indicated in the graph with the following significance codes: ‘***’ for *p* < 0.001, ‘**’ for *p* < 0.01, ‘*’ for *p* ≤ 0.05 and *p* ≥ 0.05.

**Table 1 viruses-15-01469-t001:** Summary of the characteristics and the effect of *aglB* deletion on the three infectious agents used in this study.

	HRTV-DL1	HFPV-1	PVs (pR1SE)
**Genome size**	37.7 kb	8 kb	37.8 kb (pR1SEDL18 = 109 kb)
**Particle**	head-tailed virus, non-contractile tail	pleomorphic, membrane enveloped	undescribed morphology, membrane enveloped
**Life cycle**	lytic	chronic	chronic
**Predicted glycosylated virus/plasmid proteins**	ORF12 (major capsid protein) ORF24 (host attachment) [26]	ORF4 (spike protein) [39]	ORF6 (structural protein), ORF9 (structural protein) [41]
**S-layer interactions**	binds S-layer as primary receptor [26]	not known	not known
**Glycosylation-related genes on genome**	ORF26 (predicted glycosyl hydrolase [26])	none	ORF89 (predicted glycosyltransferase/glycogen phosphorylase), ORF90 (predicted glycosyltransferase) [41] from plasmid pR1SEDL18
**Effect of host N-glycosylation on life cycle of infectious agent**	not significant	significant increase in HFPV-1 extracellular titers and intracellular virus:host genome ratios in ∆*aglB* cells	significant decrease in extracellular pR1SE/PV titers in ∆*aglB* cells
**Effect of N-glycosylation of particles on particle stability in varying NaCl concentrations**	changes in stability in varying NaCl concentrations are similar to parent strain	particle stability of ∆*aglB*-produced particles does not increase when comparing low-salt to physiological salt conditions, as observed for particles produced in the parent strain, indicating altered particle stability	not tested

## Data Availability

Not applicable.

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
