# Peer review of "Influence of N-Glycosylation on Virus–Host Interactions in Halorubrum lacusprofundi"

_viruses, 2023, doi:10.3390/v15071469_

Round 1

Reviewer 1 Report

General:

Gebhard et al, studied the impact of N-glycosylation on infection of viruses and plasmid versicles of Halorubrum. This is an interesting topic and the authors have done their best to get more insight into this unexplored area.

Major comments

·       I have trouble with the claimed differences between the AglB mutant and wt as shown in Figure 4 and 5. At first I was confused, because I expected a logarithmic scale on the y axis. However, this is not the case, and the differences are not even one order of magnitude. For a virus titre, that is very little. As virion production is impacted by many different factors, natural variation in titers can already be 1 order of magnitude. Thus the error bars in this experiment are also very large. Is this difference really something relevant? Or is it just because the nr of repetitions was not yet high enough? Would the difference disappear when you perform the experiment 20 times?

·       It would be interesting to know if the particles produced by parental and aglB deletion strain show the same specific infectivity

·       “3.2 The life cycle of the lytic virus HRTV-DL1 is not influenced by changes to the host N-288 glycosylation pathway” - That the aglB deletion has no effect on the infection with an arTV is not particularly exciting and could also be put in the Supplementary.

·       Why was the stability of the produced particles only tested against NaCl concentrations and not also against temperatures and pH?

·       The discussion is very long and the last part on plasmid copy numbers (in Sulfolobus) does not seem very relevant. Please shorten it to keep the focus.

Minor comments:

Line 19                Both, (comma).

Line 19                To what is the term ‘’infection efficiency’’ referring to, in this context? The “Efficiency of infection” can usually be calculated and expressed as a percentage value.

Line 80 - 81        Do the viruses encode glycosyltransferases? (Only mentioned for the PVs).

Line 95 – 98       In this sentence it is not completely clear if HFPV-1 or HRPV-1 was previously shown `                to be glycosylated in a host depended manner.

Line 99                How can be discriminated between PVs and EVs. I’m missing a short explanation.

Line 109              In what media was the wild type ACAM34_UNSW strain grown then?

Line 112             Sensitivity rather than adaptability? Particles do not adapt to changing concentrations they are sensitive to low/high salt.

Line 113             NaCl concentrations. No other salts tested. What are the concentrations of other salts in the DBCM2+ medium? What sodium chloride concentrations were tested?

Line 137              (0.2% [w/v])

Line 139              Biological or technical replicates, or both?

Line 157 + Line 166  + 179           “Cell Synchronization”: The cell cycle of haloarchaea has not yet been studied and there is no evidence that the cells can be synchronized (apart from using methods as flow cytometry and cell sorting). Until evidence is presented that the cells do indeed progress synchronously through their cell cycle, this formulation should not be used. Moreover, the sense of such a "synchronization" is not clear to me.

Line 164              “diluted into fresh media” Which media?

Line 164              grown until OD600 reached approx. 0.8

Line 165              OD600 of 0.05

Line 166              OD600 of 0.05

Line 168              4,500 x g (spacing), temperature?

Line 171              centrifugation temperature?

Line 176 – 177   Reference or prove that this ratio is sufficient/effective?

Line 180              4,500 x g (spacing), temperature?

Line 182 ff         Why are the free virus particles in the supernatant not removed? If they later quantify the ratio in the supernatant vs the pellet, the amount in the supernatant contains the initial unbound particles as well?

Line 188             The centrifugation force should be given in g (rpm is device-specific). Centrifugation temperature? What type of table centrifuge? Manufacturer?

Line 189 + 194  What medium?

Line 193 – 197 I don't understand why this extended cultivation with several dilution steps is necessary.

Line 202             virus, (comma)

Line 204             Gncs (start of the sentence: upper case)

Line 216             Upper and lower case?

Line 218             dye (lower case)

Line 221             HRTV-DL1, HFPV-1, or pR1SE plasmid

Line 224             g/l, 180 g/l, or 240 g/l NaCl. (comma)

Line 226             Here it says only the WT strain was infected, while in Line 228 the MOIs for parental (pyrE knockout strain or WT?) and aglB knockout strains are listed.

Line 228             Why are the infections carried out with different MOIs? And MOIs that are not far apart. Couldn't they just be adapted to each other? This would make the experiment much more comparable and no subsequent normalization would be necessary.

Line 229 - 231   For comparability, the same host to cell ratio should be used for infection. It makes sense that there are differences in yield between the individual viruses and PVs. But why for the same virus/PVs different MOIs or ratios are used for infection is not clear to me. Control and deletion mutant should be exposed to the same amount of particles for comparison.

Line 232             I doubt that the normalization against the lower MOI is accurate. As the MOI increases, so does the proportion of cells infected with at least one viral particle. So, if infected with different MOIs, the number of infected cells will also be different and thus the genome copy number per ml. The experiment should either be carried out with the same conditions, or it should be shown/proven that the proportion of infected cells is the same with an MOI of 1 and 5 or 3.5 and 5.3 ect. (or: for normalization, the factor in which the proportion of infected cells increases with different MOI should be shown).

Line 246 ff         The results of the growth curve in Figure 1 do not correspond to the results of the supplementary Figure 1. In the supplementary figure, the growth of the aglB KO strains at 180 g/l NaCl corresponds to the parental strain (no differences). In the main text, however, it is shown that the knockout reaches higher ODs.

                            Then it is claimed that growth in aglB knockouts is negatively affected at high and low salt concentrations. I can recognize this for 240 g/l  (Figure1) and 250 g/l (Figure S1) but not for the lower NaCl concentrations. Here the Figure 1 shows a impacted growth of the aglB KO at 120 g/l NaCl. In the Supplementary Figure1, however, the growth of parental strain and aglB KO seems to be affected equally.

Line 256             The aglB strain is said to be immotile, but Figure 1C indicates a halo formation of 0.5 diameter (with a slight increase on day 12). Was the inoculation point measured here? A larger image of the halo formations in Figure 1C would help to identify the halos. This is jut too small. When you want to make comparison between strains, they have to be both inoculated on the same plate (here this seems not the case, or is it?). Also, it seems that no technical replicates were made. This should be done.

Line 260 – 264  As already mentioned, the same thing is shown here and in Figure S1. Only with slight differences in the low and high NaCl concentrations but different effects shown in Figure 1 and Supplement 1.

Line 266             Biological or technical replicates, or both?

Line 274             Is there any Quantification from the micrographs?

Line 275 ff         It is very difficult to see this in the shown TEM images, also because the images are very small. Can other TEM images be shown? That better correspond with the description in the text (potentially cryo-EM? To increase resolution )

Line 284 – 285  Biological or technical replicates, or both?

Line 306             Is this the p value average for all three samples/time points samples were analyzed?

Line 323             Supplementary Figure2: It is not easy for me to see the glycoprotein stain in Fig2A.  The same applies to the Coomassie stain in all three pictures. But especially 3A.

Line 328             Here it is speculated about the effect the aglB deletion has on the infection efficiency of particles produced by the KO strain, but no further prove that statement is given.

Line 333             As for the other graphs as well, the rise of and the differences in titers would be more obvious if a different axis scale is chosen.

Line 338             (6.42*10+8 ± 4.8*10+7, n=3) and (1.89*10+8 ± 2.4*10+7 (spacing)

Line 338             What kind of statistical test was chosen here? A T-test does not seem correct to me. The titers are very similar and are in the range of 10^8. The difference can hardly be that significant. Effect is only significant for one time point (44 hpi).

Line 342             Figure 4A Different scaleing for Titer, Figure 4B: with the time scale on the x-axis it appears like samples of two strains were taken at different time points (after each other).

Line 358             Mention in the Material and Methods the kind of statistical test that was chosen.

Line 360             1.37*10+6 *ng DNA-1 (±2*10+5,n=3) (+before upper case for consistency)

Line 361             (±3.9*10=5, n=3) (+before upper case for consistency)

Line 365             Indicate statistical differences in the Graphs (by Asterix or something similar).

Line 365             Is that not to be expected? The particles produced accumulate in the supernatant.

Line 374             Figure 5:

The order of the letters in the figure legend is not alphabetical.

                            Figure 5B: With the labelling of the x-axis it is not possible to see which samples of parental strain and knockout strain were taken at the same time and can be compared with each other. What about the times when no graph can be seen? Are the levels there too low to be displayed?

Figure 5C: Why does the measurement of aglB deletion strain on the right stops earlier?

Line 389             It was not shown that the cell envelope is instable. Maybe better formulated to “the effect of aglB knockout”.

Line 403             It was only shown that it sensitive to low salt concentrations.

Line 405             NaCl concentrations.

Line 410             Indicate statistical significance in the figure. Name the kind of statistical test in the Material and Methods

Line 412             “The infectivity of the ΔaglB strain produced virions that remained slightly below the parental control, regardless of the condition.” I would recommend to reformulate this sentence. It’s hard to understand.

Line 423             “salt concentrations” instead of “conditions”

Line 436             Figure 6: See above about correction against different MOIs used for infection - As the MOI increases, the percentage of cells infected with at least one viral particle also increases - higher MOI + more infected cell = higher gncs

Line 453             I doubt this since Figure1 and Figure S1 show differences in normal and low salt growth.

Line 472             “Salt concentration has a significant effect on the stability of virus parti-cles produced in both strains.” Seems wrong statement in a row saying effect of aglB deletion (and not salt concentration).

Line 482             Only low salt with effect.

Line 490 and Line 492    I do not see this (experiments about adsorption, viral binding, uptake or cell recognition) shown/performed in this paper.

                             “N-glycosylation of the host 491 receptor, the S-layer glycoprotein, does not affect host recognition, virus adsorption and 492 uptake into the cell” – sentence shoulb be removed.

Line 495             “The host range arTVs of the Caudoviricetes class…” – this formulation makes no sense.

Line 532 ff         Very speculative. Only  results about aglB deletion are shown, not anything about the effects on s-layer stability. I find this conclusion to much.

Line 601             Speculation. Where has increased entry been demonstrated?

Line 505             That release particles likely by budding– should be toned down, as not proven.

Line 544             Likely depended on the host glycosylation machinery. (not Proven)

Line 564             What does the (0.04) mean?

Author Response

Our response has been uploaded as pdf file.

Reviewer 2 Report

The paper describes the impact of N-glycosylated proteins on the stability and the infection of Halorubrum lacusprofundi with two different virus particles, the head-tailed HRTV-DL1 and the membrane-containing HFPV-1, and also of plasmid vesicles transferring plasmid pR1SE. The viral particles and also the plasmid membrane vesicles (PVs) were instable and less infective when derived from a N-glycosylation negative mutant of Hr. lacusprofundi and present in high salt media. The authors discuss the implications of their findings and promote a possible intermediate position of PVs between plasmids and true viruses. The paper contains valuable information on the subject and also compares the results obtained with other archaeal virus-host systems.

Specific comments and questions:

Title: the paper only deals only with haloarchaeal viruses and plasmids and not with different archaeal viruses. This should be specified.

Introduction:

It should be explained why the impact of N-glycosylation was analyzed with Hrr. lacusprofundi and the Haloferax volcanii virus HFPV-1 instead of Hfx. volcanii. The virus is less infective on the cells of Halorubrum. Just to compare this membrane-surrounded virus particle with the head-tailed Hrr. virus HRTV-DL1? It might be nice to have a table on the different haloarchaeal viruses mentioned in the introduction. What are the genome sizes and do they carry genes involved in glycosylation? (The Table on these viruses found in the discussion is too late).

Line 179-181: Synchronization of cells:

The original paper on synchronization of bacterial cells used cells in the early stationary phase – please explain why you harvested the cells at mid-exponential phase.

Are the cells “synchronized by serial solution”? Cells are not in “solution” but are in “suspension”. Is it possible to perform a serial solution? The term “serial dilutions” is often used, serial solutions? The same problem is with the PV “solution” – PVs do not dissolve, they are present in suspension.

L 185: is a temperature of 28°C the optimal growth temperature of H. lacusprofundi? Please specify.

L199, 202,204: what are plasmid gcns? Please explain when this term is used first. Are these genes?

L215: SDS-PAGE: Please mention the buffer system of your SDS-PAGE.

Figure 1C: the photographs of the motility assay are very small and hard to see – please move to the right side of the graph and enlarge the two photos.

L290: what is arTV? Please explain.

L294: It should be made clear that ORF24 is the product of the respective ORF, i.e. a protein. Otherwise the sentence is confusing to read.

L315: Normalized to the level of the “main host chromosome”: are there also minor host chromosomes in this strain?

L392-393: Sentence is unclear – are words missing?

Table 1: the information of the first part of Table 1 is required much earlier in the paper (see above)

L569, 576: In addition to the “main chromosome”: are there “minor chromosomes”?

Mega-plasmids of haloarchaea: the mega-plasmids of H. salinarum are “mini-chromosomes”, since they contain essential genes. Thus, these are not plasmids or “mega-plasmids”.

L575: Euryarchaeida: I never heard this term. They should be Euryarchaeota

L585: non-native Sulfolobus strains: what is this? Appears to be lab jargon.

L595: What is meant by “the edge of the viral frontier”

look in my report

Author Response

(The authors gave the same response as above.)

Reviewer 3 Report

The submission by Gebhard et al. reports the deletion of a putative glycosyl transferase gene of Halorubrum lacusprofundi and the resulting impact on infection of this extremely halophilic archaeon with viruses and naturally produced plasmid-containing vesicles.  The functional significance of the glycosylation of cell-surface proteins of archaea remains an important question, and use of gene deletions and biological assays are suitable approaches for its study.  The experimental methods of the study generally are described in appropriate detail, and the literature references are more than adequate; in fact, they could be made fewer and more selective.

The Introduction provides no coherent overview of the basic biochemistry of protein glycosylation; topics that would seem important to address in this section might include the features that are most broadly conserved across evolution, the major stages of the process, and the basic types of glycosyltation (O- vs. N-linked, for example).  Existing evidence for the protein glycosylation pathways possessed by this specific organism also is not presented systematically, making it difficult to evaluate the rationale and implications of the choice of gene that was deleted.  

The manuscript interprets several experimental results in terms of the stability of the host cell envelope.  Although this is a plausible explanation, the study did not measure cell-envelope properties by independent biochemical or physical assays.  Thus, envelope stability remains an hypothesis, not a molecular mechanism demonstrated to be the most relevant cause of observed effects.  This point must be clarified to help avoid misinterpretation by some readers.

The lack of a unified overview of the organism's predicted glycosylation capabilities is compounded by the lack of any confirmation of how the gene deletion affected the actual glycoprotein composition of the host cells.  This removes a critical link in the logic of the study, preventing observed biological effects from being associated with a demonstrated change in the glycosylation of one or more cell components.  Remarkably, no glycoprotein analysis is presented even for virus or plasmid-containing particles produced by mutant cells, despite its being done for particles from wild-type host (Suppl. Fig. 2).

The following comments refer to the indicated line of the manuscript.

Title
The title sounds like a broad literature review rather than a focussed experimental study, and thus is misleading.  A simple correction would be to replace "Archaea" with "the Archaeon Halorubrum lacusprofundi" or something similar.

86
A paragraph should be inserted here which summarizes the available evidence regarding glycosylation pathways of this organism and any of its cell-surface glycoproteins that have been detected or characterized.

140
The abbreviation "EV" should be spelled out and a literature reference provided.

199
The abbreviation "gcns" is cryptic, and since the term appears in only a few places, it would be most helpful simply to spell it out each time.

216
The biochemical basis of this assay should be identified, so that readers do not have to find and decipher the company's literature in order to know how it works.

290
"arTV" should be spelled out.

306
This is an unnecessarily awkward sentence which needs re-organization.

330
How can HFPV-1, or any virus that infects a micro-organism, be 'chronic'?  Please re-state in biologically definite terms what is intended here.

385
The logic of this claim is not clear.  If the plasmid encodes its own glycosyl transferases, it seems plausible that the infected cell would be less dependent, rather than more dependent, on host-encoded enzymes.

488
The evidence presented in the manuscript does not exclude the possibility of other enzymes being able to N-glycosylate proteins in the absence of functional aglB.

503
Given the fundamental cytological differences between archaea and Gram-positive bacteria, the archaeal S-layer perhaps may be described as analogous to peptidoglycan, but not truly equivalent to it.

521
What alternative mechanism do the authors propose that would not be determined by the virus and would not involve altering the normal S-layer structure?

527
Since the impact of the aglB deletion on protein glycosylation was not measured, it remains unknown whether this conclusion is valid, and to which host proteins it may apply.

541, 557
What is the glycoprotein composition of virus particles and PVs produced by the aglB mutant?

560-595
This section is a tedious, wandering discourse that would improve the focus of the manuscript by being removed.  Any relevance to the rest of the study should be explained clearly in one or two sentences.

600
Where did the study prove directly that the aglB mutant makes no N-glycoprotein?

The quality of English is good, with few exceptions that have been noted in other comments.

Author Response

(The authors gave the same response as above.)

Round 2

Reviewer 1 Report

General:

In its entirety, the manuscript appears too hastily and quickly revised. Some comments and remarks from the first revision were not addressed or changes were not implemented consistently. Some new features such as the Coomassie and glycostaining gels have not improved at all.

It is not sufficiently clear from the Material & Methods how often experiments were repeated and whether replicates and error bars represent repeated measurements of the same type of organisms treated or grown in the same conditions several times (biological variation), or whether the experiment was performed once with repeated measurements of a sample (variation of equipment and or protocols)..

Line 30

Is the major outcome that further experiments are necessary?

Line 44

Repetitive sentence. Neither for N-glycosylation nor for O/glycosylation is much known.

Line 74

The Slayer is not an cell appendage.

Line 80

The official ICTV naming of the virus does not require the letter R to be capitalized. (Halorubrum Pleomorphic virus 6).

Line 77 – 107

Here, the difference between crenarchaeal viruses and haloarchaeal viruses as well as basic differences in virus morphologies should be better elaborated (e.g. arTVs compared to membrane lipid containing pleomorphic viruses). The paragraph jumps back and forth between crenarchaeal viruses and haloarchaeal viruses and for a reader outside the field it is not clear what basic differences exist between the host organisms and the potential infection mechanisms of their viruses. Furthermore, the S-layers and their glycosylation of haloarchaeon and crenarchaea are different.

Line 99

“The majority of archaeal viruses are thought to utilize the host N-glycosylation machinery for glycosylation of their own proteins.”

I think this is only true for pleomorphic viruses and some Crenarchaeal viruses? At least the chosen citations refer only to Halorubrum pleomorphic virus 1 (HRPV-1) – for some tailed viruses it was also suggested that they use the host glycosylation machinery to alter the hosts surface for superinfection immunity.

Line 114

“Accordingly, infection efficiencies and virus particle stability of two different viruses and a virus-like element were then compared between thein parent and N-glycosylation mutant Hrr. lacusprofundi strains.”

The sentence makes no sense. The infection efficiency of the particles was tested in WT and mutant. In addition, the stability of the particles produced by the mutant and WT in different NaCl concentrations was investigated. Different things are mixed together here.

Line 138

What is Vitamin 10?

Line 143 and respective Figures as well as main text

For easier comparison with already published studies, it would be useful to indicate the NaCl concentrations in their molarity.

Line 171

Are these biological or technical replicas? Were three samples taken from one experiment or was the protease treatment carried out several times?

Line 202

Was the experiment performed several times or were only several samples taken from one infection experiment? Are they true biological replicates or just technical ones?

Line 226
Why did the ratio changed now? Was the experiment performed again?

Line 232

In your reply my first comments the authors write that "The removal of unbound PVs after infection appears to be important when analyzing PV abundance in the supernatant of the culture". Why do Material & Methods still claim that the particles in the supernatant were not removed? Was the experiment repeated?

Line 294 – 303

How many replicates were performed of these experiments? If performed once with 3 samples taken from the same infected culture, the statistics give only accuracy about measuring the genome copy number.

Line 294 ff

Again, infection efficiency depends on the MOI - different MOIs can lead to differences in the particles produced. An MOI of 1 is less efficient than an MOI of 10 and experiments with such differences should not be compared. Otherwise, experiments should be provided showing that infection experiments with different MOIs result in the same infection efficiencies.

It may be that the receptors are saturated at high MOI, so a higher number does not necessarily mean more infections. The frequency of receptors is not known. Therefore, a more appropriate comparison would be with the same MOIs or I would like to see the infection efficiency with different MOIs.

Line 300

“The impact of the strain on infection efficiency” Should be formulated to something like “The influence of host strain glycosylation (as a particle producer) on particle stability in NaCl” to make clear what is meant here.

Line 306

“S-layer instability”

See comment below: In my opinion the instability is insufficiently proven. The S-layer might only be easier accessible by the protease.

Line 312

In contrast to the citation (18) for H. volcanii, where the instability of the S-layer glycoprotein-derived glycopeptides was determined by MALDI TOF analysis, quantification of protein levels after trypsin digestion and cryo-EM analysis of the slayer in WT and mutant, here we cannot speak of "accordingly".

Quantification of protein concentrations after protease treatment by comparison with a house keeping protein or via Bradford assay would be appropriate. The statement about the stability of the S-layer is based solely on protease degradation. In my opinion, this experiment does not provide sufficient evidence that the S-layer is more unstable as a result. Especially since the mutants show the same growth rate as the WT in all infection experiments and the growth curves presented in the supplementary, and a destable S-layer is likely to have disadvantages for cell physiology. It could also be that the lower glycosylation simply makes the S-layer more accessible to the protease. In my opinion, further experiments are needed to support this statement. For example, MS analysis of the S-layer protein, or cryo-EM or at least some kind of quantification of increases release of S-layer proteins to the medium.

Figure 1B

Font size to small

Line 315 ff and Figure 1C and Supplementary Figure 2

How often was the growth curve measurement carried out?

Line 361

Why no confirmation of protein species by Mass Spectrometry?

Table 1

The wording of the last column of the table (as well as the main text) leads to confusion. If we are talking here about the general infections efficiency of the particles, this must be clearly distinguished from the stability in NaCl and the infections efficiency after incubation in different NaCl concentrations. Also later in the main text, stability of particles in NaCl is equated with infection efficacy, which is not easy to understand. perhaps infection efficacy after NaCl treatment or similar would be better.

Furthermore, statistically non-significant results should not be mentioned as an effect. If the test shows that differences are not significant, they do not need to be listed.

Figure 3

I do not recognize any of the Asterix mentioned in the legend in the Figure.

Line 428/Figure 4

“Statistical significance is indicated in the graph using the same significance codes as described above.”

 Please insert information here and do not refer to other figures.

Line 460 + 461

A statement about particle stability cannot be made at this point. How can it be excluded that the particles not only have difficulties in leaving the host cells?

Line 509 + Line 512

“and slightly .reduced infection efficiency in high-salt” + “were more stable in high-salt 512 conditions.”

As already mentioned: If a result is not statically relevant, it should not be presented as an effect or result of the tested conditions. After all, the statistical analysis has shown that it is only a matter of natural variance.

Line 513

“The salt concentration itself did not have a statistically significant effect on infection efficiencies of particles produced in the ΔaglB strain. Nevertheless, infection efficiency was reduced under optimal and high-salt conditions, supporting our hypothesis that when compared to particles from the parent strain. This strongly indicates that HFPV-1 particle stability is negatively impacted when N-glycosylation is impaired.”

Again: if not statistical significant this cannot be stated.

Line 518

I only see a significant effect for low salt conditions and not "varying" conditions.

Line 520

“indicating that the deletion of aglB negatively affected PV particle stability as it did for the two viruses.”

There was no significant effect for the Pleolipovirus. Needs to be reformulated.

Line 522

“Infection efficiency of PVs produced in the ΔaglB strain was in general significantly higher (p=0.005** at 120 g/l and p=0.009** at 180 g/l)”.

Only significant differences in low and normal salt. Not “in general”. Reformulate.

Line 529/Figure 6

“Figure 6: Deletion of aglB negatively affects HFPV-1 and PV infection efficiencies”

Stability or infection efficiency/infectivity?

A: high and low salt show higher infectivity than normal salt - so how is there a huge difference in infectivity - misinterpretation of statistics.

Please indicate in the Figure legend and/or Material and Methods if the stability experiments were performed in independent biological replicates or not.

Discussion

The discussion is still very long.

Line 548 and 522 ff

A statement about the effects of N-glycosylation on cell stability cannot be made for the reasons already mentioned unless further experiments are made available to substantiate this claim.

Line 568

I don't feel that statistically irrelevant results need discussion.

Line 575

No effects could be detected in HPFV1. And for the other virus, effects in high salt were not significant - The sentence is generally repetitive and should be in results.

Line 603

Because of lower ratios

Line 607

“destabilization of the cell envelope caused by 607 incorporation of non-N-glycosylated S-layer glycoproteins”

Speculation.

Conclusion

Line 681

“Destabilization of the host cell envelope, specifically the S-layer, leads to increased entry 681 and exit of membrane-enveloped HFPV-1, indicating that HFPV-1 might depend on 682 naturally occurring gaps in the S-layer for exit and entry.”

Speculation.

Others

Author Response

The point to point response is attached as pdf file.

Reviewer 3 Report

Extensive revision has addressed many concerns of the reviewers and strengthened the manuscript significantly.  On some points, however, information addressing the reviewer's concerns was not added to the manuscript, and thus not communicated adequately to the reader.  These omissions, described below with respect to the 'clean copy' line numbers, should be straightforward to correct.

109
It would seem appropriate and helpful to paraphrase the following response to reviewer 3, preferably with the predicted functions of some of the genes, and incorporate it somewhere in this first paragraph:
" . . .little is known about the N-glycosylation pathway in Hrr.
lacusprofundi, although a putative glycosylation cluster was identified (Hlac_1062 to 1075, Kaminski et al 2013, https://doi.org/10.1016/j.ympev.2013.03.024) . . ."

286
Some readers may find the gels difficult to interpret; perhaps the major S-layer subunit bands could be marked with small arrowheads or something similar to help readers find and compare them.

307/540
The following rationale (made in response to a reviewer comment), should be included, with corresponding literature citations, in the first paragraph of section 3.1 or of section 4:
"To date, every archaeal species deleted of aglB has shown a loss of N-glycosylation (except for Sulfolobus acidocaldarius, where aglB is essential and could not be deleted). In the case of Haloferax volcanii, evidence for the existence of a second oligosaccharide that is used in the appearance of a second N-linked glycan generated in low-salt conditions was presented (i.e., Kaminski et al., 2013 https://doi.org/10.1128/mBio.00716). Even there, however, the absence of AglB compromised N-glycosylation."

The following response is valid and accurate should be paraphrased and incorporated somewhere the first paragraph of section 4: ". . . we do not show that the aglB mutant is not capable of any N-glycosylation, but we show that N-glycosylation is impacted (e.g. Supplementary Figure 1). . . "  

In contrast, the following statements that seem to imply a complete absence of glycoprotein was confirmed remain in the revision and should be removed or re-worded more precisely and consistently with the limitations of the data (i.e., Suppl. Fig. 1):

494 "Deletion of aglB prevents N-glycosylation in Hrr. . . ."
516 " . . . lack of glycosylation . . .
528 " . . . lack of N-glycosylation of virus proteins . . .

617
Insert the genus & species to clarify that the results of the study apply to Hrr. lacusprofundii cells.

Author Response

(The authors gave the same response as above.)
